# The modeling and condition analysis of nondestructive testing based on ESPI for internal defects of materials

**Wen Wang**[1,2], **Mingzhi Lv**[2,3], **Fang Zhang** ![ORCID][2,4]*, **Zhitao Xiao**[2,4], **Wenheng Li**[3,2]

**1** School of Mechanical Engineering, Tiangong University, Tianjin, China, **2** Tianjin Key Laboratory of Optoelectronic Detection Technology and Systems, Tianjin, China, **3** School of Electronics and Information Engineering, Tiangong University, Tianjin, China, **4** School of Life Sciences, Tiangong University, Tianjin, China

\* hhzhangfang@126.com

## Abstract

Electronic speckle pattern interferometry (ESPI) is a non-contact, full field, real-time measurement technology, which judges the position and size of the internal defects of the object through the external deformation caused by the internal defects under certain loading conditions. We present the effect of loading mode and loading parameters to the defect detection. Firstly, the finite element analysis method is used to establish models to simulate the defect detection of aluminum plates under different loading conditions. Mechanical models are established to simulate different loading mode, loading sizes, defect depth and defect sizes. Secondly, the interpolation method based on partial differential equation is applied to obtain the whole field out-of-plane displacement after finite element analysis. Thirdly, by analyzing the interference fringe patterns obtained from the out-of-plane displacement caused by different defects, the deformation rules in the detection of internal defects of aluminum plates are obtained under different loading conditions. Finally, the loading mode and loading range suitable for the internal defect detection of aluminum materials are summarized. This method can provide a basis for the selection of loading mode and parameters in the ESPI experimental system.

## 1 Introduction

Manufacturing is an important support for economic and social development. Defects may occur during the manufacturing and use of workpieces. The surface and internal defects of the workpiece include shrinkage defects, porosity defects, inclusion defects, and so on. Among them, holes and inclusions are the most common internal defects of aluminum die castings. When the workpiece is stressed, it will cause the stress concentration on the defect, which will lead to the decline of the mechanical properties and the irreversible damage of the workpiece. This damage will greatly

Data availability statement: All relevant data are within the manuscript and its Supporting Information files.

Funding: This work was supported by Tianjin Metrology Technology Project (2024TJMT037).

Competing interests: The authors have declared that no competing interests exist.

reduce the service life of the material and have a great impact on safety production in serious cases. Therefore, the influence of defects on workpiece structure and physical properties cannot be ignored. Compared with surface defects, internal defects are more hidden. It is crucial to detect possible internal defects of workpiece [1]. At present, nondestructive testing (NDT) technologies for workpiece defects mainly include traditional radiographic testing (RT), ultrasonic testing (UT), penetration testing (PT) and eddy current testing (ECT).

With the development of modern science and technology, electronic speckle pattern interferometry (ESPI), an effective method to compensate for traditional NDT, appeared in the 1970s. Compared with conventional methods (RT, UT, ECT), ESPI offers unique advantages in specific scenarios:1. Real-time full-field monitoring: ESPI can provide a global view of the surface deformation of the workpiece, and analyze the dynamic changes in strain caused by defects under loads (such as mechanical or thermal stress) in real time, which is superior to the point scanning mode of UT/ECT. 2. Non-contact and non-destructive: No physical probes or coupling agents are required as in the UT method, nor is there the radiation risk as in the RT method, making it safer for operators.3. High sensitivity: detecting micron-scale displacement, revealing subtle defects or residual stresses that may not significantly change the internal structure but affect the surface behavior, good at detecting near-surface defects, complementary to RT/UT. It plays a huge role in the detection of aerospace, biomedical, mechanical and other fields. For example, Retheesh [2] et al. studied the defects in low modulus materials used as insulators in solid rocket motors by using laser speckle interferometry, and obtained that when the thermal load reaches the temperature difference of 20°C to 28°C, the compound will produce the best stripe pattern for defect analysis. This study demonstrates that the simulation and prediction of the amount of thermal loading is of great significance for improving the detection sensitivity. Lin [3] et al. used ESPI to examine the visible displacement fringes in order to elucidate the anti-phase as well as in-phase motions associated with vibration. Li [4] et al. used amplitude-fluctuation electronic speckle pattern interferometry (AF-ESPI) to obtain the mode shapes at different resonant frequencies, so that to identify the presence of the waviness defects in carbon fibre reinforced plates.

When ESPI technology is used for qualitative detection of materials, the defects can be detected as long as the material surface has a small bulge of the wavelength level under appropriate loading condition [5]. However, due to the diverse types, sizes, depth of defects, materials of workpieces and other factors, it often requires repeated tests to find the appropriate loading conditions, mostly relying on the experience of operators. Therefore, exploring the loading conditions under different test pieces and defect factors is an important guarantee for accurate ESPI measurement.

In order to find appropriate experimental conditions for ESPI, people have made many attempts to improve the accuracy and effectiveness of internal defect detection. The finite element analysis is a commonly used modeling method for ESPI. Yang [6] et al. combined with the principle of digital shearing speckle interferometer, established a model using finite element analysis to simulate the defect detection of aluminum plates and composite laminates under different load conditions, verified that

the defects shown through thermal loading and vacuum loading are easily to be recognized. But the detection capabilities of different loading amounts for defects of different sizes and depths were not specifically evaluated. Diaz-Mendoza [7] et al. give a numerical analysis of the residual stress distribution in a three-point bending test of a TRIP sheet by using finite element based on ESPI. The results prove that the finite element simulation calculation results have a good correlation with the experimental results.

The main contributions of this work are as follows:

(1) The out-of-plane displacement of the defect position after vacuum loading and thermal loading is simulated by FEA, and then the phase is calculated by the displacement to obtain the ESPI stripe. By studying the detection ability of different loading amounts for defects with different diameters and depths, an important basis can be provided for how to select loading mode and parameter in ESPI nondestructive testing and evaluate the defect detection capability of this testing technology.

(2) The full-field displacement values are obtained by applying the partial differential equation (PDE)-based interpolation method to the discrete displacement data from the finite element mesh simulation. This PDE-based approach enables accurate interpolation of the displacement field across the entire domain.

In the following sections, we describe the principles of ESPI measurement and simulation in detail in Part 2. In Part 3, we analyze the influence of two loading mode on the test results, namely vacuum loading and thermal loading, and give the experimental results and discussion. Part 4 concludes the contents of this study.

## 2 Principles of measurement and simulation

When heating radiation or negative air pressure is applied to an object, the internal defects of the object will change differently from other parts of the object, thus affecting the surface deformation of the object. By analyzing the change of speckle field on the surface of an object, we can detect whether there are defects inside the object. When testing workpieces with different materials, structures and defects, the conditions such as loading time, loading mode and loading intensity have crucial impact on the testing effect [8]. Because the defect is inside the object and the shape variable is small, if the loading time is too long or too short, or if the loading mode or loading intensity is improper, it will increase the difficulty of defect detection, and the detection results are affected, and it may even cause missed defects detection.

To solve this problem, finite element analysis software ABAQUS is used to simulate aluminum workpiece and its preset defects, and the interference fringes formed by the speckle field on the workpiece surface are generated by Matlab software. Based on the simulated results, the shape variables generated by different defects under different loading conditions are analyzed from a theoretical point of view, which can help us to find the appropriate loading conditions.

### 2.1 Principle of electronic speckle pattern interferometry

Electronic speckle pattern interferometry (ESPI) is a non-contact and high-precision deformation measurement method. It uses laser speckle as the information carrier of deformation displacement [9] to measure the deformation of rough surfaces, and then infers whether there are defects inside the workpiece.

As shown in Fig 1, a coherent beam is divided into reference light and object light through the beam splitter. The object light projects on the surface of the tested workpiece, and the formed reflected light interferes with the reference light beam which directly projects from the laser to the camera. The interfered result of the two beams forms a third speckle field, which is recorded by the camera.

Two speckle images before and after the object deformation are recorded respectively. Then the interference fringe pattern is obtained by digital subtraction of the two speckle images. The visible fringes that cannot be observed in a single light field can be obtained by subtraction, as in formula (1) [10].

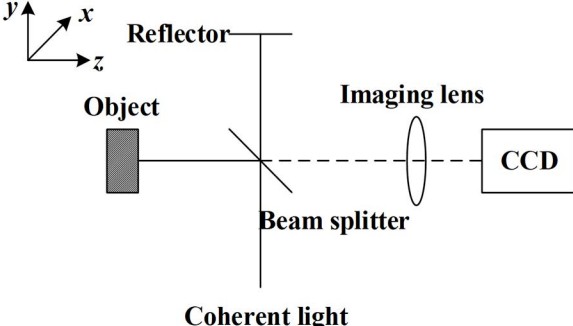

**Fig 1. Electronic speckle interference optical path system.**

$$I_{sub}(x,y) = \left| I_{before} - I_{after} \right| = 4u_O u_R \left| \sin\left[ \delta(x,y) + \varphi(x,y)/2 \right] \sin\varphi(x,y)/2 \right| \tag{1}$$

where $I_{before}$ and $I_{after}$ are the corresponding interference light intensity before and after object deformation, $u_O$ and $u_R$ represent the amplitude of object light and reference light. The high frequency term $\sin\left[ \delta(x,y) + \varphi(x,y)/2 \right]$, is related to the random fluctuation of object surface and corresponds to speckle noise. The low-frequency term $\sin\varphi(x,y)/2$ is only related to the deformation of the object surface and corresponds to the interference fringes. When $\sin\varphi(x,y) = 0$, that is $\varphi(x,y) = 2k\pi$, a minimum value of dark fringe patterns will appear; When $\sin\varphi(x,y) = 1$, that is $\varphi(x,y) = (2k+1)\pi$, the maximum value of bright fringe patterns will appear.

By analyzing the speckle movement or change recorded on the speckle pattern, we can not only obtain the displacement and strain information of the object surface, but also judge whether there is a defect in the object interior.

For the optical path for out-of-plane displacement measurement shown in Fig 1, the relationship between the displacement on the object surface and the relative phase difference is [11]:

$$\varphi = \frac{2\pi}{\lambda}(2\omega) \tag{2}$$

where, $\omega$ is the displacement component along the $z$ direction, namely the out-of-plane displacement. Therefore, when the laser wavelength is fixed, the range of out-of- plane displacement of the object can be controlled by controlling the range of the load applied to the object to obtain the phase difference within a certain range.

## 2.2 Simulation principle

Finite element analysis can simulate the real physical system (geometry size and load cases) by using mathematical approximation method. The continuous structure is discretized into finite elements, in each of which finite nodes are set, and the continuum is regarded as a collection of a group of single elements connected only at the nodes [12].It is an effective tool to transform a finite degree of freedom problem in a continuous domain into a finite degree of freedom problem in a discrete domain.

In this study, the vacuum loading and thermal loading simulation are conducted for the defects inside the aluminum plate to explore the most appropriate loading intensity for defects with different depth and sizes. A thin square plate with a side length of 10 cm and a thickness of 5 mm is designed using ABAQUS, and a circular defect is made in the middle of the plate. The distance from the top of the defect to the upper surface of the plate is $H$, and the diameter of the defect is $D$. The material properties used for modeling are shown in Table 1 and the model is shown in Fig 2. When the test piece

**Table 1. Material characteristics of aluminum plate modeling.**

| Material characteristic | Value |
|---|---|
| Density (kg/m³) | 2700 |
| Young's modulus (GPa) | 70 |
| Poisson's ratio | 0.3 |
| Conductivity (W/(m·K)) | 237 |
| Coefficient of expansion (1/°C) | 2.36E-05 |
| Specific heat capacity (J/(kg·°C)) | 879 |

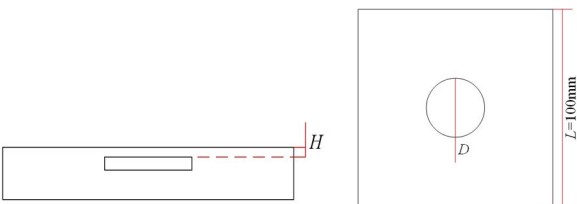

**Fig 2. Schematic diagram of model: (a)** Side view of model and **(b)** Front view of model.

is heated or the pressure of the environment is less than the atmospheric pressure, the expansion of the internal defects of the test piece will cause the out-of-plane displacement of the test piece surface. We analyze the above deformation information through finite element simulation, and the specific process is as follows.

(1) The deformation of the specimen after being loaded is simulated by ABAQUS finite element analysis software, and the out-of-plane displacement of each grid point of the specimen is further obtained (as shown in Fig 3).

(2) The out-of-plane displacement information of each point on the upper surface of the test piece derived from ABAQUS is imported into the data analysis software Matlab (as shown in Fig 4(a)), and the discrete displacement data obtained from finite element analysis is interpolated using the partial differential equation (PDE) image processing method [13] to obtain the displacement distribution at the interval of pixel steps (as shown in Fig 4(b)).

(3) Assuming the laser wavelength is 633nm, the phase corresponding to the out-of-plane displacement is calculated using Eq. (2), and then the fringe patterns are generated. The visualization result of fringe patterns is shown in Fig 6, where (a) is the fringe pattern corresponding to the displacement before interpolation, and (b) is the fringe pattern corresponding to the displacement after interpolation).

(4) The value of loading conditions required for defect detection of aluminum plate can be determined according to the clarity of the fringe pattern.

Figs 4(a) and 5(a) shows the surface displacement $\omega_0$ of the specimen simulated by ABAQUS finite element analysis software. Because there are a finite number of nodes on the grid during the finite element analysis, the out-of-plane displacement of the aluminum plate surface is only obtained at the grid nodes. In order to obtain the displacement distribution $\omega$ which is approximately continuous in vision and is spaced by pixel step size, the phase interpolation method based on partial differential equation (PDE) is used to process the displacement data $\omega_0$. For the convenience of describing the principle, the nodes with known displacement during finite element analysis are called the known nodes, and the nodes requiring interpolation to obtain the displacement are called the unknown nodes.

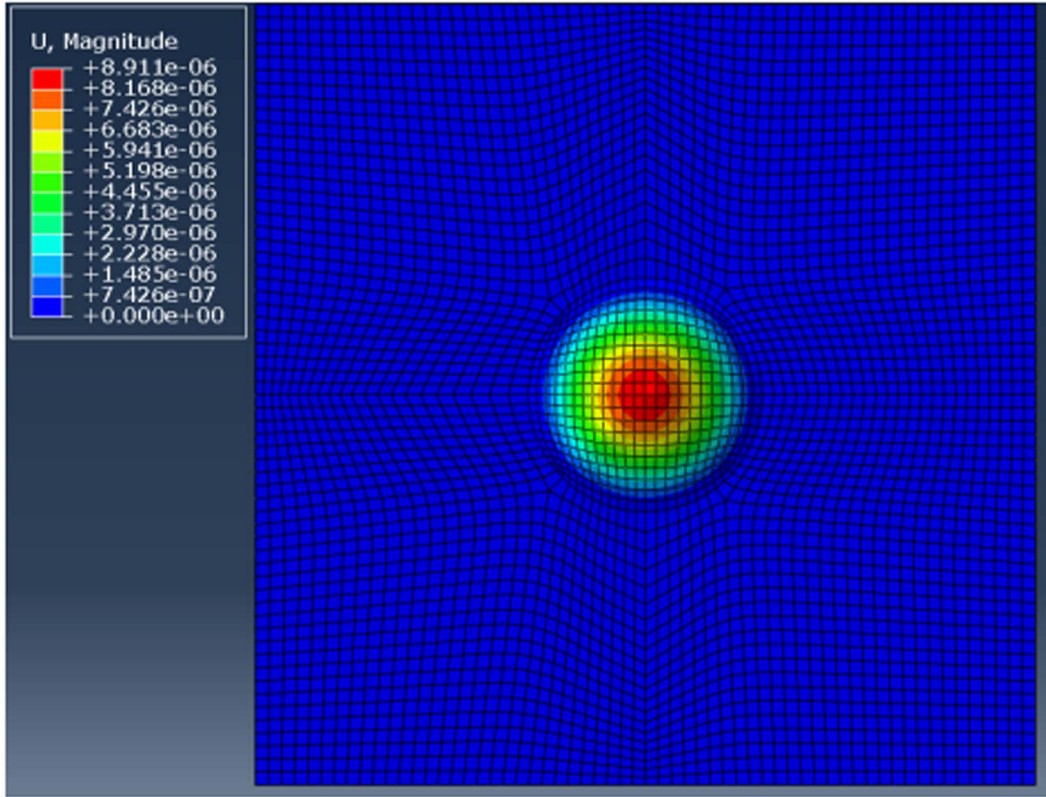

**Fig 3. Workpiece deformation diagram and the out-of-plane displacement.**

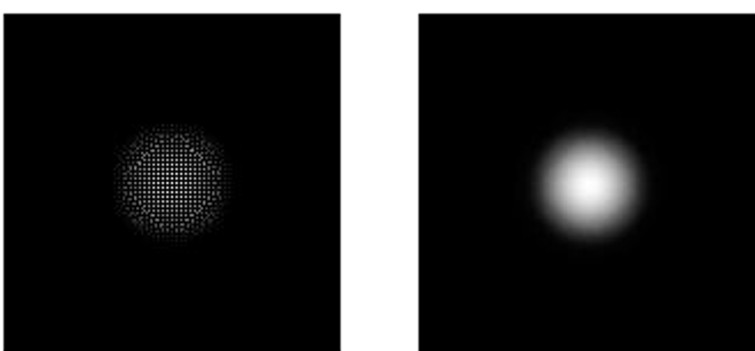

**Fig 4. Two-dimensional view of the full-field displacement: (a)** Displacement map before interpolation and **(b)** Displacement map after interpolation.

According to the principle of heat conduction, the full-field displacement value of the aluminum plate surface can be regarded as the temperature distribution. The displacement value of each known node can be regarded as the initial temperature value, and the temperature of the unknown node is zero, as shown in Fig 4(a). According to the principle of heat conduction, the temperature value on each node provides heat, and the heat will be transferred from the place with high temperature to the place with low temperature until the relative balance is reached. Therefore, the temperature value on the high temperature node will change the temperature value outside the node. The temperature value of the whole field

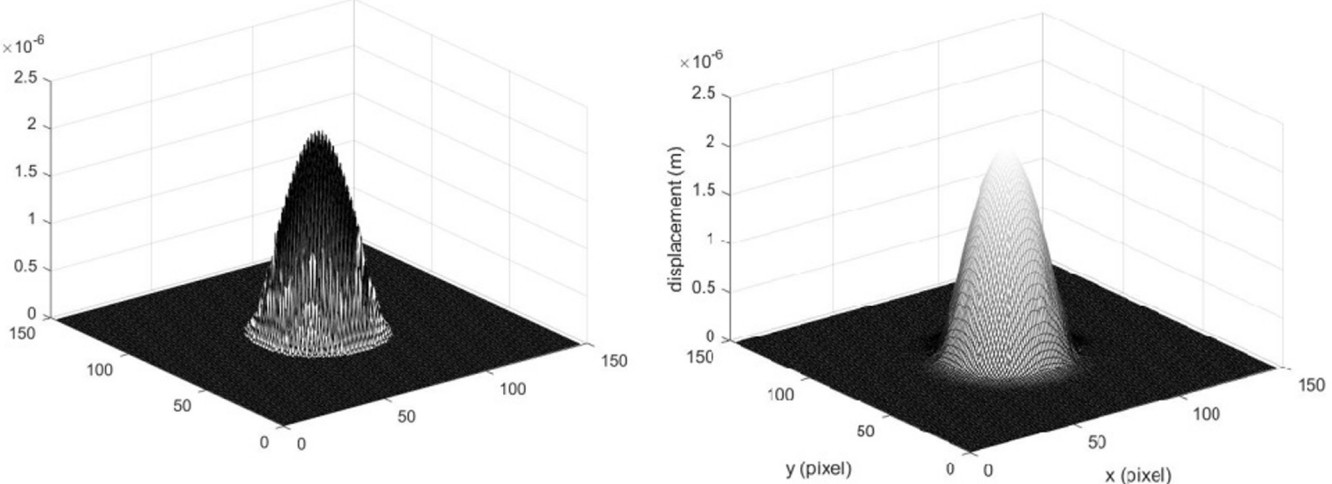

**Fig 5. Three-dimensional view of the full-field displacement: (a)** Displacement map before interpolation and **(b)** Displacement map after interpolation.

will undergo heat exchange, and the temperature will be redistributed and reach a stable state after a period of time, so as to achieve the purpose of displacement interpolation and obtain the whole field displacement.

The principle of heat conduction is the core idea of PDE interpolation, and the equation is shown in equation (3).

$$\partial_t \omega = \nabla^2 \omega \tag{3}$$

where $\omega(x,y,t)$ is the temperature value of each node after a period of heat conduction, that is, the displacement after interpolation. The initial conditions are as follows:

$$\omega(x, y, 0) = \omega_0(x, y) \tag{4}$$

$\omega_0(x,y)$ is the initial displacement of each known node. The above process is achieved by iteratively solving the solutions of partial differential equations. Energy is conserved, so in the process of thermal diffusion, the temperature value (i.e., displacement) will decrease with the diffusion of energy and eventually deviate from the true value. Therefore, energy correction is required. Before each iteration, the displacement value on the known node is reassigned to the initial value, which can not only provide continuous energy for the temperature change of each point outside the known node, but also ensure that the displacement value on the known node remains unchanged. The conditions for energy correction are as follows:

$$\omega(x_s, y_s, t_n) = \omega_0(x_s, y_s), (x_s, y_s) \in S \tag{5}$$

where $S$ is the set of known nodes, and $n$ is the number of iterations when solving the equation iteratively. According to the above initial conditions and energy correction conditions, the interpolated results of the heat conduction equation can be obtained, as shown in Fig 4(b) and Fig 5(b).

In the electronic speckle pattern interferometry, the displacement information of the object is hidden in the fringe pattern of the speckle pattern interferometry, that is, the fringe pattern is the basis for judging whether there are defects in the

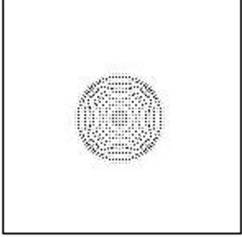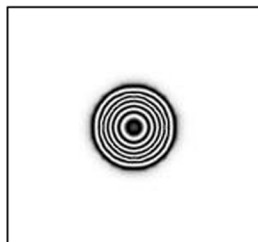

**Fig 6. Fringe pattern corresponding to the full-field displacement in** Fig 4: (a) before interpolation and (b) after interpolation.

specimen and for quantitative analysis of the defects. Therefore, the fringe pattern is generated for the interpolated displacement data (Fig 4(b)) according to Eq. (6), as shown in Fig 6(b).

$$I(x, y) = \cos\left[\varphi(x, y)\right] = \cos\left[\frac{4\pi\omega(x, y)}{\lambda}\right]$$

(6)

where the laser wavelength is assumed to be 633 nm, which is a common choice for ESPI technology. According to the density of the obtained fringe pattern, we can judge whether the loading amount set for a defect is appropriate, and then explore the most suitable loading amount for defect detection, providing guidance for the actual ESPI defect detection.

## 3 Loading mode and parameter analysis

We analyze the influence of two loading mode on the test results, namely vacuum loading and thermal loading. The vacuum loading is conducted by putting the sample into the sealing box. The air in the sealing box is extracted by the vacuum pump to make the air inside the defect expand, that is, the negative pressure is applied to the surface of the test piece. Under the loading of vacuum negative pressure, defects will be subjected to uniform load, which results in deformation of defect surface [14].Thermal loading is using heating devices to heat the test piece, so that different parts of the test piece will expand accordingly. Defects are different from other parts of the test piece when heated, thus showing different surface deformation. Thermal loading has the advantages of fast, real-time, simple operation, non-contact, etc. The test object in our simulation is aluminum plate, and aluminum has high thermal conductivity and thermal expansion coefficient, which is easy to produce surface deformation after heating. Defects with large diameters ranging from 20.0 mm to 60.0 mm and small diameters ranging from 5 mm to 10.0 mm were simulated. This type of defect often occurs in the manufacturing of large components (such as ships and Bridges) or in the damage of complex service environments (such as deep sea and high temperature). The depths were all within 1 mm to verify the detection force of ESPI technology for near-surface damage.

### 3.1 Vacuum loading

The principle of vacuum loading is the same as that of pressure loading. When using ABAQUS for modeling, the defect part will deform by peripheral fixation and loading on the defect part for the test piece. The loading intensity is the size of negative pressure applied on the defect surface. The deformation data are recorded under different loading intensity for different defect depth and different defect diameters, and the fringe patterns are displayed by Matlab. The simulation results are shown from Tables 2–5.

 **3.1.1 Effect of load intensity.**  The load intensity has an important influence on the observable fringe pattern at the defect location. When the load is too small, the surface deformation of the test piece caused by the expansion of internal defects is small, and the fringe patterns are too sparse or cannot produce complete fringe pattern, which makes it difficult to identify defects from the sparse fringe patterns. On the contrary, if the load is too strong, the surface of the test piece

**Table 2. Maximum out-of-plane displacement and fringe pattern under different loading.**

| Loading capacity (pa) | Defect diameter $D$ (mm) | Defect depth $H$ (mm) | Maximum out-of-plane displacement $\omega$(m) | Fringe pattern |
|---|---|---|---|---|
| 100 | 40 | 0.5 | 3.85E-07 | Fig 7(a) |
| 200 | 40 | 0.5 | 7.70E-07 | Fig 7(b) |
| 300 | 40 | 0.5 | 1.16E-06 | Fig 7(c) |
| 500 | 40 | 0.5 | 1.90E-06 | Fig 7(d) |
| 1000 | 40 | 0.5 | 3.85E-06 | Fig 7(e) |
| 2000 | 40 | 0.5 | 7.53E-06 | Fig 7(f) |

**Table 3. Out-of-plane displacement and fringe patterns with different defect depth (I).**

| Loading capacity (pa) | Defect diameter $D$ (mm) | Defect depth $H$ (mm) | Maximum out-of-plane displacement $\omega$(m) | Fringe pattern |
|---|---|---|---|---|
| 500 | 40 | 0.3 | 3.07E-06 | Fig 8(a) |
| 500 | 40 | 0.4 | 1.87E-06 | Fig 8(b) |
| 500 | 40 | 0.6 | 1.16E-06 | Fig 8(c) |
| 500 | 40 | 0.7 | 7.40E-07 | Fig 8(d) |
| 500 | 40 | 0.8 | 4.03E-07 | Fig 8(e) |
| 500 | 40 | 0.9 | 2.93E-07 | Fig 8(f) |

**Table 4. Out-of-plane displacement and fringe patterns with different defect depth (II).**

| Loading capacity (pa) | Defect diameter $D$ (mm) | Defect depth $H$ (mm) | Maximum out-of-plane displacement $\omega$(m) | Fringe pattern |
|---|---|---|---|---|
| 1000 | 40 | 0.5 | 3.85E-06 | Fig 9(a) |
| 1000 | 40 | 0.6 | 2.31E-06 | Fig 9(b) |
| 1000 | 40 | 0.7 | 1.17E-06 | Fig 9(c) |
| 1000 | 40 | 0.8 | 8.06E-07 | Fig 9(d) |
| 1000 | 40 | 0.9 | 5.86E-07 | Fig 9(e) |
| 1000 | 40 | 1.0 | 4.33E-07 | Fig 9(f) |

**Table 5. Maximum out-of-plane displacement and fringe patterns under different defect diameters.**

| Loading capacity (pa) | Defect diameter $D$ (mm) | Defect depth $H$ (mm) | Maximum out-of-plane displacement $\omega$(m) | Fringe pattern |
|---|---|---|---|---|
| 500 | 20 | 0.5 | 1.30E-07 | Fig 10(a) |
| 500 | 30 | 0.5 | 6.01E-07 | Fig 10(b) |
| 500 | 40 | 0.5 | 1.88E-06 | Fig 10(c) |
| 500 | 50 | 0.5 | 4.53E-06 | Fig 10(d) |
| 500 | 60 | 0.5 | 9.32E-06 | Fig 10(e) |

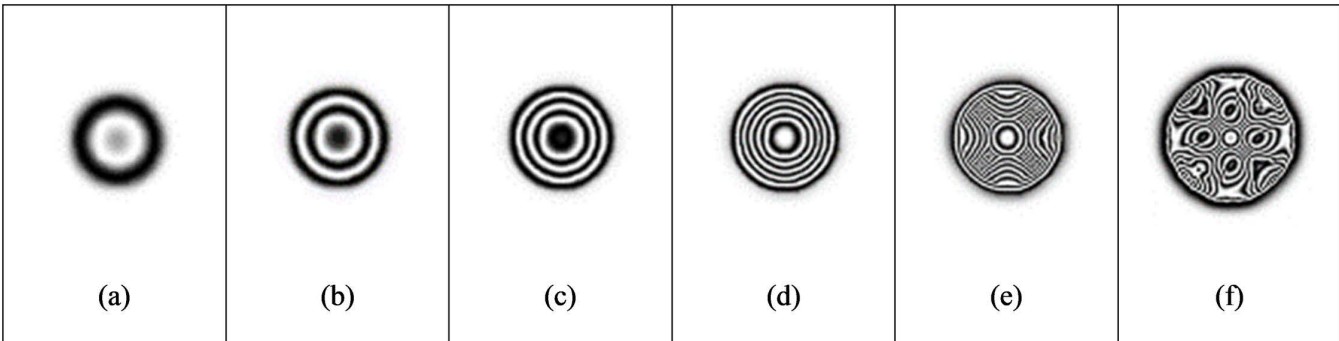

**Fig 7. Fringe pattern under different loading.**

will deform too much, resulting in too dense interference fringes and difficult to observe. Firstly, the influence of load intensity on the test results is studied when the defect diameter is 40 mm and the defect depth is 0.5 mm. From Table 2, it can be concluded that when the load is 100pa, the interference fringes begin to form. The fringe density increases with the increase of load. According to the principle of phase measurement, the fringe density is directly proportional to the displacement gradient. When the load intensity increases, the displacement gradient on the material surface increases, resulting in a decrease in the spacing between adjacent interference fringes, thereby increasing the fringe density (Fig 7).

When the load reaches 1000pa, the out-of-plane displacement is close to 4 μm. The fringes are dense, and the adjacent fringes interfere and overlap with each other. When the load is increased to 2000pa, the surface deformation of the object is too large, the fringes are too dense, and the size and shape of the fringe area are different from the actual defects, so effective interference fringe signals cannot be obtained. Comparing Table 3 with Table 4, under the condition of the same defect diameter and defect depth, the loading intensity has a greater impact on the out-of-plane displacement (Figs 8 and 9).

**3.1.2 Effect of defect size.** For nondestructive testing, the minimum size of detectable defects is an important indicator of testing means. As shown in Table 5, the fixed defect depth is 0.5 mm. With the increase of the circular defect diameter, the density of fringe patterns gradually increases under the same loading intensity of 500 pa (Fig 10).When the defect diameter is 20 mm, under the loading condition of 500pa, the effective interference fringes cannot be obtained due to the small out-of-plane displacement, so the defect cannot be detected. The fringe density increases with the increase of defect size. The fringe density is proportional to the displacement gradient. The fringe density is proportional to the displacement gradient. When the loading amount is constant and the defect diameter increases, it will be easier

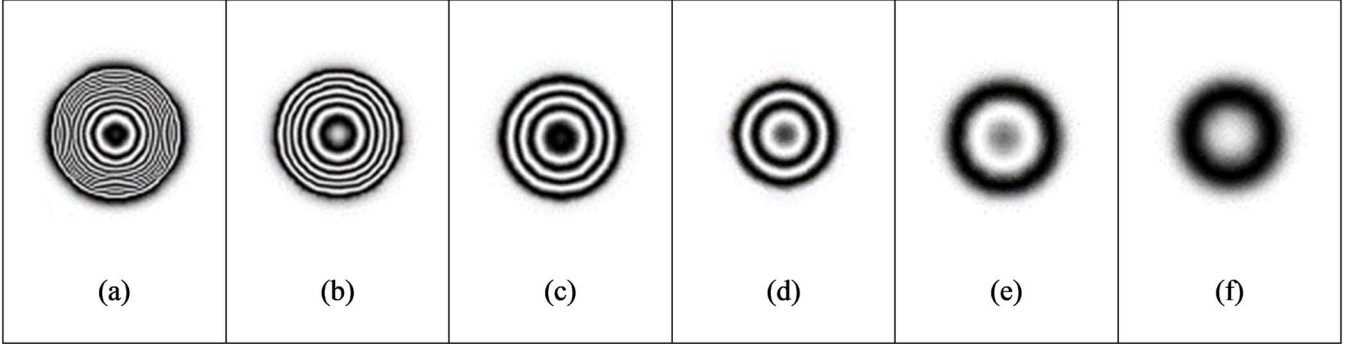

**Fig 8. Fringe patterns with different defect depth (I).**

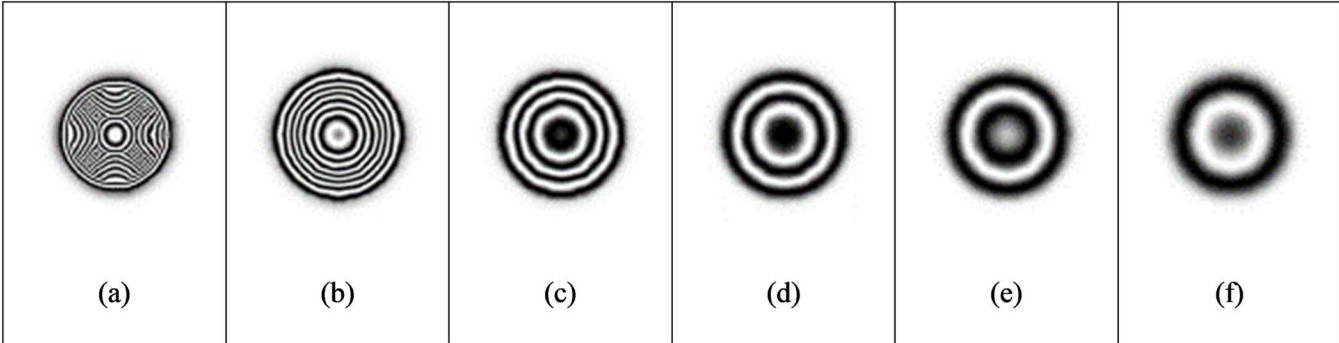

**Fig 9. Fringe patterns with different defect depth (II).**

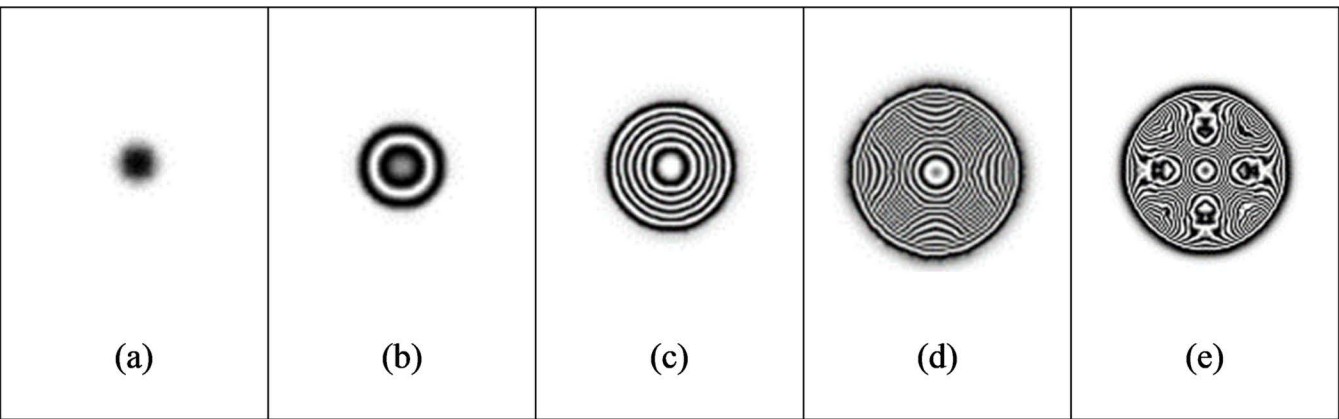

**Fig 10. Fringe patterns under different defect diameters.**

to produce out-of-plane displacement, which will increase the fringe density. When the defect diameter is increased to 60 mm, the fringe patterns are too dense to appear aliasing, and the effective interference fringes corresponding to the defects cannot be observed. Therefore, according to the results of finite element simulation, for the internal defects 0.5 mm away from the upper surface of the aluminum plate, the defects with a diameter of less than 30 mm will be difficult to be detected when the vacuum load is 500pa.When the fixed defect depth is unchanged, the larger the defect diameter is, the smaller the loading intensity required to obtain a clear fringe pattern is.

**3.1.3 Effect of defect depth.** Detectable defect depth is another important indicator of nondestructive testing technology. In Table 3, the fixed defect diameter H is 40 mm, the defect depth is increased from 0.3 mm to 0.9 mm, the loading intensity is 500pa, and the fringe patterns are gradually sparse, that is, the deeper the defect is, the harder it is to be detected. At this time, the loading intensity needs to be increased to detect clear fringe patterns. For the aluminum plate with an internal defect diameter of 40 mm, when the vacuum loading is 500pa and the defect depth is greater than 0.9 mm, the defect is difficult to be detected. Similar patterns can be seen from Table 4.

From this point of view, it is especially important to find the appropriate loading intensity for different specifications of defects in the defect detection process. Using the simulation model to evaluate the conditions required for defect detection in advance and finding the appropriate loading scheme and loading intensity have an important guiding role for actual defect detection.

## 3.2 Thermal loading

With the help of ABAQUS finite element analysis software, the sequential coupled thermal stress analysis method [15] is used for the heat conduction analysis first, and then the analysis results are imported into the thermal stress analysis module to complete the transient thermal stress coupling analysis. A 400 mm × 500 mm × 5 mm solid aluminum plate is built, whose back is drilled some round hole defects with different diameters. The diameters ranging of these holes is from 1 mm to 20 mm and the diameter changing step is 1 mm, as shown in Fig 11. The distance between each two holes is large enough to not affect each other. The distance between the upper surface of the fixed defect and the observation surface is 0.5 mm.

Taking a single hole as the research object, in ABAQUS finite element simulation, establish a size of 100 mm × 100 mm × 5 mm aluminum plate for modeling and simulation, and the heat flow density is used to simulate the process of thermal loading. In the heating process, the increase of heating time will increase the deformation of the test piece, which will affect the test results. In addition, the deformation of the object is too fast during heating, which is not conducive to the detection of the deformation of the defect. Therefore, the object is heated first and then cooled during thermal loading. During the cooling process, the deformation of the test piece is recorded and compared with the test piece before heating, record the shape variable data and present the fringe pattern using Matlab.

The change of defect and fringe pattern in different defect diameters, different heat flux, and different heating time were investigated, respectively. The halogen lamp is usually used as the heat source in the existing hot loaded ESPI defect detection. In the simulation, the heating method of equivalent heat flux is adopted, which can be expressed by Eq. (7):

$$q = P/S \tag{7}$$

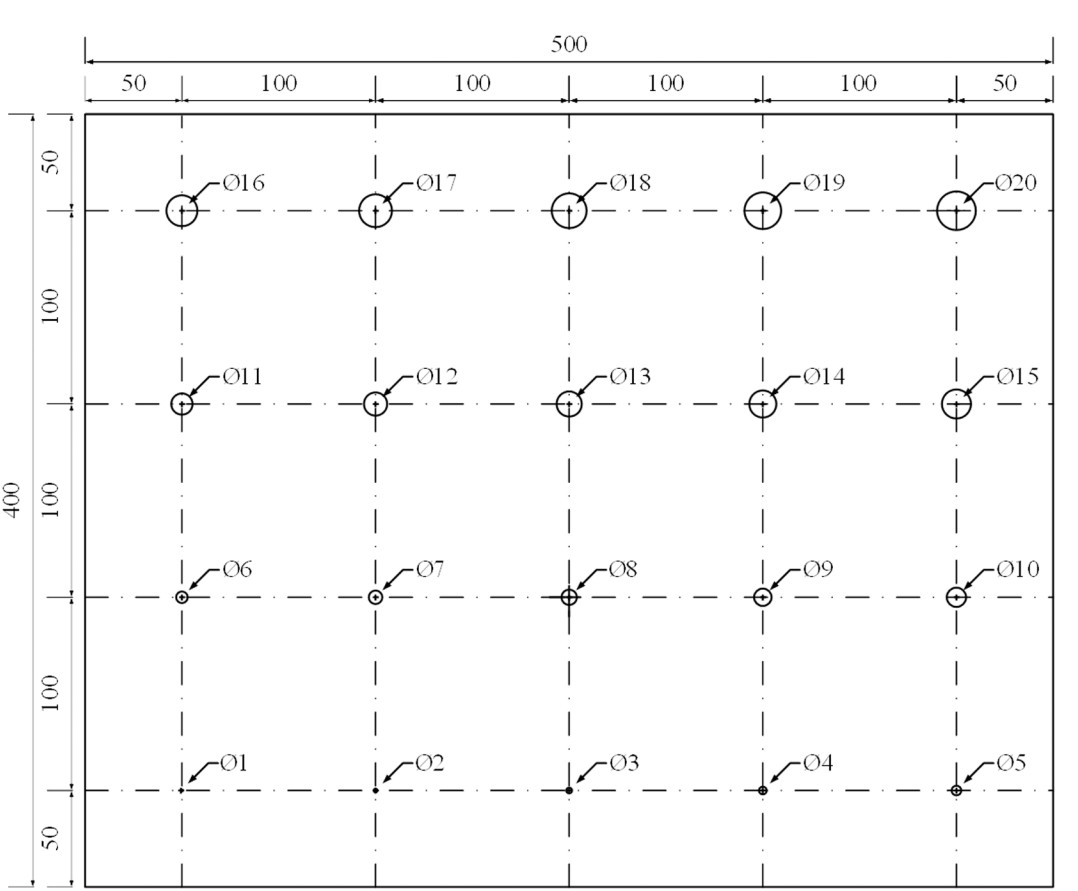

**Fig 11. Structural dimension of the aluminum plate.**

where $q$ is the heat passing through the unit area in unit time, $P$ is the heat flow rate (power), and $S$ is the sectional area. The power of halogen lamp is 500W, 1000W, 1500W and 2000W. The energy loss will be about 50% during the detection. The maximum area during the detection is 0.2 m². Therefore, the equivalent heat flux is calculated as 1250W/m², 2500W/m², 3750W/m² and 5000W/m².

**3.2.1 Effect of defect size.** The distance between the upper surface of the fixed defect and the observation surface is 0.5 mm, the heating time is 3s, and the cooling time is 50s. We add 2500 W/m² of heat flow density to the upper surface of the aluminum plate, and change the size of the defect diameter $D$. The maximum out-of-plane displacement and fringe pattern of defects with different diameters are explored. It can be seen from Table 6 that under the same conditions, with the reduction of the defect diameter, the defects are more difficult to be detected. Interference fringes can be detected when the defect diameter is 10 mm or more, while when the defect diameter is less than 8 mm, it is difficult to be identified effectively because the defect diameter is too small.

Fig 12 gives the maximum out-of-plane displacement of defect diameter diagram under different heat flux, which shows the comprehensive effects of defect size and loading strength on out-of-plane displacement (Fig 13).

**3.2.2 Effect of load intensity.** Keep the defect depth of 0.5 mm, heating time of 3s, and cooling time of 50s, and set the heat flow density of 1250 W/m², 3750 W/m², and 5000 W/m² respectively, as shown in Tables 7–8, and Table 9 (Figs 14–16). Select the defect diameter of 5 mm, 10 mm, 15 mm, and 20 mm for comparison. It is found in the process of exploration that the greater the load intensity, the greater the maximum out-of-plane displacement of defects under the same conditions. Taking the defect diameter of 20 mm as an example, when the heat flow density is 1250W/m², the fringe pattern initially appears, and the maximum out-of-plane displacement of the test piece is about 4e-07m. With the increase of heat flow density, the fringes become increasingly dense. When the heat flow density is 5000W/m², the fringes are very dense and adjacent fringes interfere with each other and overlap, so clear and effective interference fringes cannot be observed.

**3.2.3 Effect of heating time.** The effect of defect detection can also be affected by adjusting the heating time. In the simulation experiment, defects with a diameter of 10 mm and a height of 0.5 mm to the upper surface are used, and the heat flow density is 5000 W/m². The heating time is increased from 1s to 5 s, and the time step length is 1s. It can be seen from Table 10 that the maximum out-of-plane displacement of defect deformation will increase with the increase of the heating time, and the definition of fringe patterns will change accordingly (Fig 17). Meanwhile, the increase of the heating time will increase the overall deformation of the test piece because the deformation of non-defective areas will also increase. Therefore, it is especially important to find a suitable heating time, which is one of the key factors to improve the detection efficiency and fringe definition.

In conclusion, the following simulation results are obtained:

Table 6. Maximum out-of-plane displacement data and fringe pattern of defects with different diameters.

| Defect diameter (mm) | Defect depth (mm) | Heating time (s) | Cooling time (s) | Heat flow density (W/m²) | Maximum out-of-plane displacement $\omega$(m) | Fringe pattern |
|---|---|---|---|---|---|---|
| 20 | 0.5 | 3 | 50 | 2500 | 1.00E-06 | Fig 13(a) |
| 18 | 0.5 | 3 | 50 | 2500 | 8.80E-07 | Fig 13(b) |
| 16 | 0.5 | 3 | 50 | 2500 | 7.761E-07 | Fig 13(c) |
| 14 | 0.5 | 3 | 50 | 2500 | 6.50E-07 | Fig 13(d) |
| 12 | 0.5 | 3 | 50 | 2500 | 5.20E-07 | Fig 13(e) |
| 10 | 0.5 | 3 | 50 | 2500 | 4.00E-07 | Fig 13(f) |
| 8 | 0.5 | 3 | 50 | 2500 | 3.42E-07 | --- |

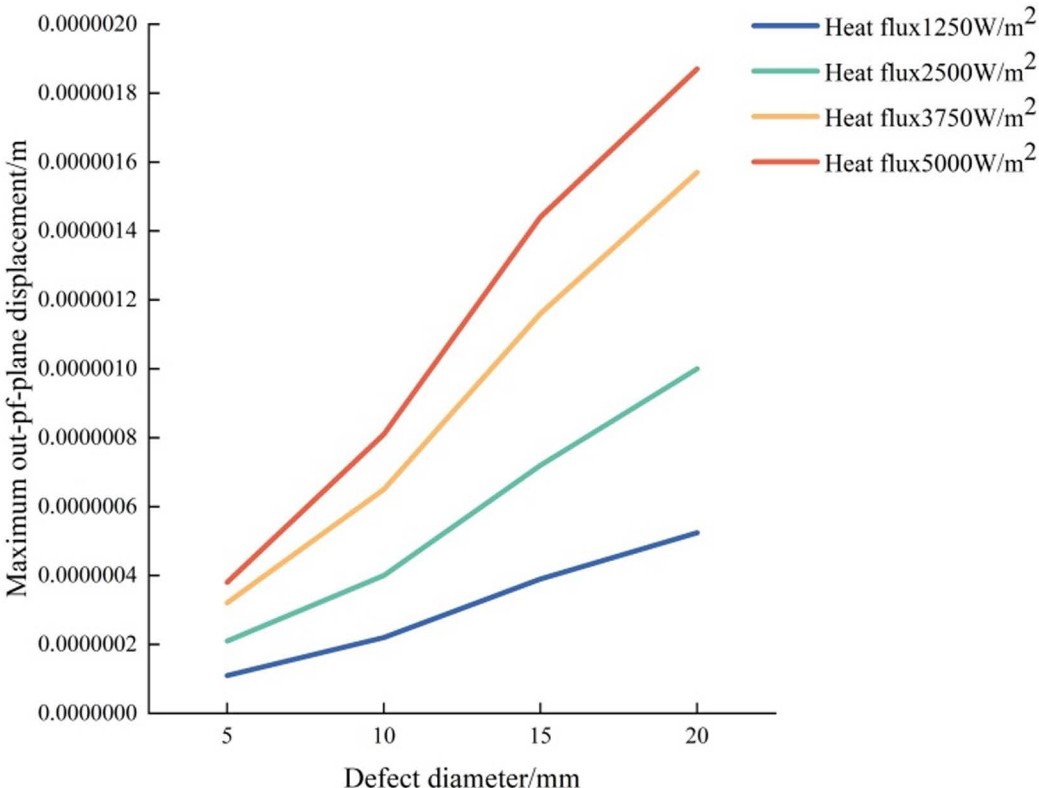

**Fig 12. Maximum out-of-plane displacement of defect diameter diagram under different heat flux.**

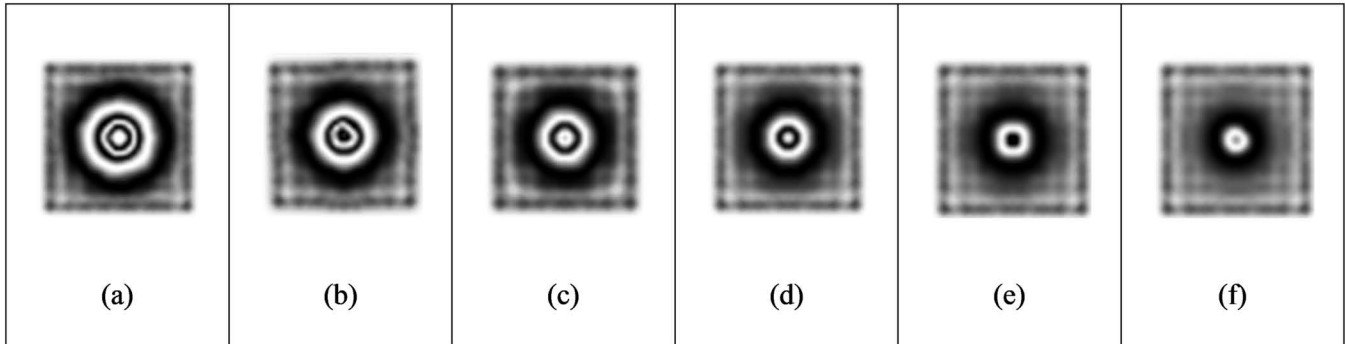

**Fig 13. Fringe pattern of defects with different diameters.**

(1) Both vacuum loading and thermal loading can be used to detect defects in aluminum plate with internal cavity defects.

(2) Under the condition of vacuum loading, when the diameter of the defect and the distance from the defect to the upper surface are fixed, the larger the loading intensity is, the denser the fringes are. The loading range of the clear fringe pattern can be displayed on the test pieces under different conditions is different. Under the same other conditions, the larger the defect diameter is, the smaller the loading intensity required to show a clear fringe pattern during detection. The greater the defect depth is, the greater the loading required to show clear fringe patterns.

**Table 7. Fringe patterns corresponding to different defect diameters when the heat flow density is 1250W/m².**

| Defect diameter (mm) | Defect depth (mm) | Heating time (s) | Cooling time (s) | Heat flux (W/m²) | Maximum out-of-plane displacement $\omega$(m) | Fringe pattern |
|---|---|---|---|---|---|---|
| 5 | 0.5 | 3 | 50 | 1250 | 1.10E-07 | Fig 14(a) |
| 10 | 0.5 | 3 | 50 | 1250 | 2.20E-07 | Fig 14(b) |
| 15 | 0.5 | 3 | 50 | 1250 | 3.90E-07 | Fig 14(c) |
| 20 | 0.5 | 3 | 50 | 1250 | 4.01E-07 | Fig 14(d) |

**Table 8. Fringe patterns corresponding to different defect diameters when the heat flow density is 3750W/m².**

| Defect diameter (mm) | Defect depth (mm) | Heating time (s) | Cooling time (s) | Heat flux (W/m²) | Maximum out-of-plane Displacement $\omega$(m) | Fringe pattern |
|---|---|---|---|---|---|---|
| 5 | 0.5 | 3 | 50 | 3750 | 3.21E-07 | Fig 15(a) |
| 10 | 0.5 | 3 | 50 | 3750 | 6.50E-07 | Fig 15(b) |
| 15 | 0.5 | 3 | 50 | 3750 | 1.16E-06 | Fig 15(c) |
| 20 | 0.5 | 3 | 50 | 3750 | 1.21E-06 | Fig 15(d) |

**Table 9. Fringe patterns corresponding to different defect diameters when the heat flow density is 5000W/m².**

| Defect diameter (mm) | Defect depth (mm) | Heating time (s) | Cooling time (s) | Heat flux (W/m²) | Maximum out-of-plane displacement (m) | Fringe pattern |
|---|---|---|---|---|---|---|
| 5 | 0.5 | 3 | 50 | 5000 | 3.80E-07 | Fig 16(a) |
| 10 | 0.5 | 3 | 50 | 5000 | 8.10E-07 | Fig 16(b) |
| 15 | 0.5 | 3 | 50 | 5000 | 1.44E-06 | Fig 16(c) |
| 20 | 0.5 | 3 | 50 | 5000 | 1.87E-06 | Fig 16(d) |

(3) Under the condition of thermal loading, when other conditions are the same, the greater the thermal load or the longer the heating time is, the denser the fringe patterns will be, and the denser the fringe patterns will be as the cooling time continues. In addition, the larger the defect diameter is, the easier the defect will be detected.

### 3.3 Qualitative analysis and prediction of defect detection based on logistic regression

Logistic regression model is a model that studies the relationship between the results of categorical variables and a set of influencing factors [16]. The purpose of this section is to establish a logistic regression prediction model for defect detection. Based on the defect features (size, depth) and the loading amount applied during defect detection, it analyzes and predicts whether a clear stripe graph will be formed, that is, to determine whether defects with certain features can be detected under specific loading conditions.

The Logistic regression model is expressed as:

$$Logit\,(P) = \ln \frac{P}{1-P} = \beta_0 + \beta_1 x_1 + \beta_2 x_2 + ... + \beta_k x_k \tag{8}$$

In the Logistic function, a linear function of parameters $\beta_0, \beta_1, ..., \beta_k$ can be generated through the Logit transformation. The Logistic model transforms the abstract prediction problem into a numerical value, where P represents the probability of the event occurring, the range of variation is from 0 to 1.

The out-of-plane displacements of defects of different specifications were obtained through FEA simulation and PDE interpolation. Based on the visualization results of Matlab, that is, whether the fringe graph is clear, it is determined

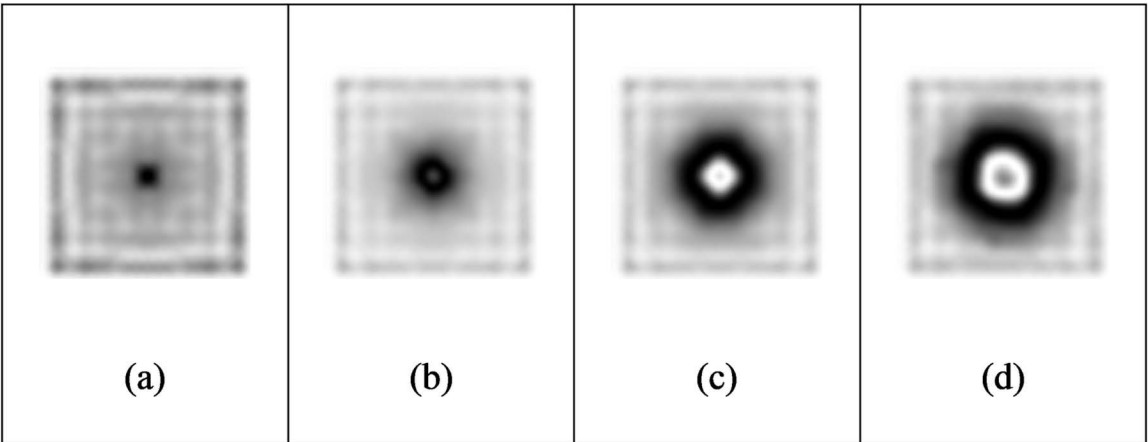

**Fig 14. Fringe patterns corresponding to different defect diameters when the heat flow density is 1250W/m².**

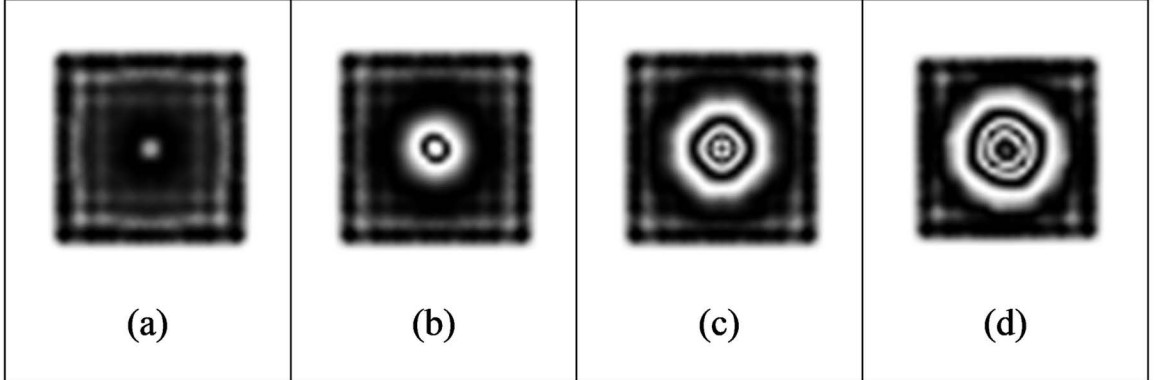

**Fig 15. Fringe patterns corresponding to different defect diameters when the heat flow density is 3750W/m².**

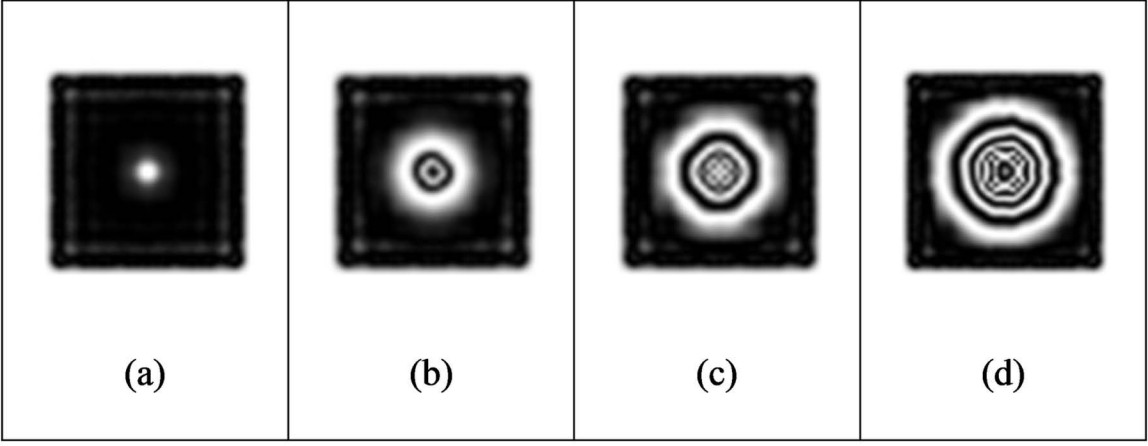

**Fig 16. Fringe patterns corresponding to different defect diameters when the heat flow density is 5000W/m².**

**Table 10. Maximum out-of-plane displacement and fringe pattern of defects under different heating time.**

| Defect diameter (mm) | Defect depth (mm) | Heating time (s) | Cooling time (s) | Heat flux (W/m²) | Maximum out-of-plane displacement (m) | Fringe pattern |
|---|---|---|---|---|---|---|
| 10 | 0.5 | 1 | 50 | 5000 | 2.70E-07 | Fig 17(a) |
| 10 | 0.5 | 2 | 50 | 5000 | 5.4E-07 | Fig 17(b) |
| 10 | 0.5 | 3 | 50 | 5000 | 8.10E-07 | Fig 17(c) |
| 10 | 0.5 | 4 | 50 | 5000 | 1.07E-06 | Fig 17(d) |
| 10 | 0.5 | 5 | 50 | 5000 | 1.34E-06 | Fig 17(e) |

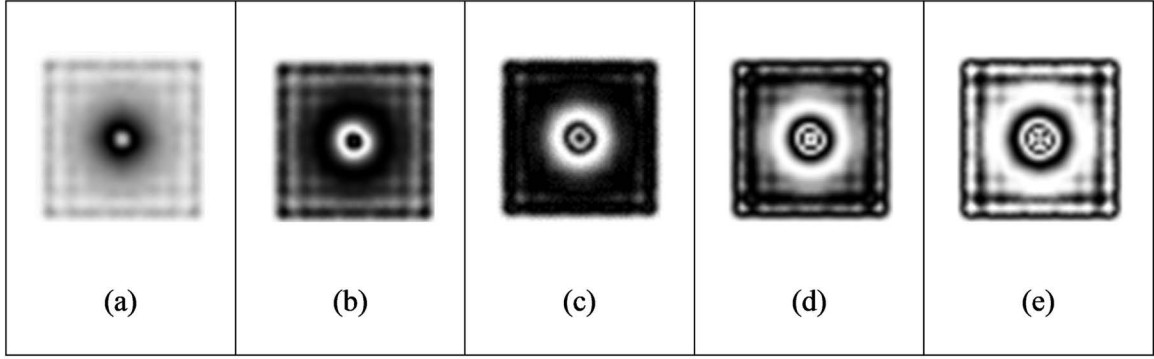

**Fig 17. Fringe pattern of defects under different heating time.**

whether each defect can be effectively detected. The research results show that the defect diameter, defect depth and loading amount can all affect the effect of defect detection. Since "whether defects are detected" is a binary categorical variable, based on the principle of the binary Logistic regression model, a binary Logistic regression model is established with "whether clear stripe patterns are detected" as the dependent variable and "defect diameter", "defect depth", and "vacuum load loading amount" as independent variables. The categorical variable "can detect clear fringe pattern" is denoted as 1, and "cannot detect clear fringe pattern" as 0.

A total of 37 samples were selected in this section for model establishment. Each sample contained data such as defect diameter, defect depth, loading amount, and maximum surface displacement. A qualitative prediction model for defect detection based on binary Logistic regression was established, and the model was applied to the prediction of another 13 samples. Since the magnitudes of each variable vary greatly, their values are normalized.

Based on the above analysis, given the quantitative values of "defect diameter", "defect depth", and "loading amount", to determine whether a clear stripe pattern can be effectively detected, the following model is constructed according to formula (8):

$$Logit\,(P) = \beta_0 + \beta_1 x_1 + \beta_2 x_2 + \beta_3 x_3 \tag{9}$$

$x_1$ represents the defect diameter, $x_2$ represents the defect depth, $x_3$ represents the loading amount, $\beta_0$ is a constant term, and $\beta_1$, $\beta_2$, $\beta_3$ are the standard coefficients.

The regression coefficient and the constant term can take either positive or negative values. When the value is negative, it indicates that the dependent variable is negatively correlated with the independent variable corresponding to the regression coefficient. It is calculated through 37 samples that the regression coefficient of $x_1$ is 3.019, the regression coefficient of $x_2$ is −4.733, the regression coefficient of $x_3$ is 7.516, and the constant term is −20.309. The defect diameter

and loading amount have a positive influence relationship on the appearance of the fringe pattern, while the defect depth has a negative influence relationship on the appearance of the clear fringe pattern. Among them, The absolute value of the regression coefficient corresponding to the loading amount is the largest, indicating that the loading amount has the greatest influence on "whether a clear fringe pattern can be detected".

The binary Logistic regression model is obtained as follows:

$$Logit\,(P) = \ln \frac{P}{1-P} = -20.309 + 3.019x_1 - 4.733x_2 + 7.516x_3 \tag{10}$$

Finally, the established binary Logistic regression prediction model was applied to predict 13 samples, and 0.5 was taken as the probability critical value to predict whether clear fringe could be detected. Table 11 shows the detailed prediction results, and Table 12 presents the statistical results.

It can be known from the prediction results shown in Tables 11 and 12 that when the established binary Logistic regression prediction model was used to predict 13 samples, the prediction accuracy rate was 92.3%.

It can be seen from Table 11 that the model makes incorrect predictions only in the samples displayed in the last row of the table. That is, when the defect diameter is 8.0 mm, the defect depth is 0.4, and the loading amount is 20,000 Pa, the predicted results are inconsistent with the observed results. The out-of-plane displacement of this situation was visualized. The corresponding fringe graph is shown in Fig 18, and the maximum out-of-plane displacement is 2.3E-04 mm. It can be found that at this time it is in a critical state where a clear fringe pattern is about to appear, that is, the model has made a prediction error only at this critical state.

**Table 11. Prediction results of the binary logistic regression model.**

| Defect diameter (mm) | Defect depth (mm) | Loading capacity (pa) | Whether there are clear fringe (observation) | Whether there are clear stripes (prediction) |
|---|---|---|---|---|
| 10.0 | 0.2 | 3000 | 1 | 1 |
| 10.0 | 0.3 | 3000 | 0 | 0 |
| 10.0 | 0.3 | 4000 | 1 | 1 |
| 10.0 | 0.4 | 4000 | 0 | 0 |
| 10.0 | 0.4 | 15000 | 1 | 1 |
| 10.0 | 0.5 | 25000 | 1 | 1 |
| 10.0 | 0.5 | 4000 | 0 | 0 |
| 10.0 | 0.5 | 1000 | 0 | 0 |
| 12.0 | 0.4 | 3000 | 0 | 0 |
| 12.0 | 0.4 | 10000 | 1 | 1 |
| 8.0 | 0.4 | 5000 | 0 | 0 |
| 8.0 | 0.4 | 40000 | 1 | 1 |
| 8.0 | 0.4 | 20000 | **0** | **1** |

**Table 12. Statistics of prediction results.**

| Actual measurement | Prediction | |
|---|---|---|
| | No clear fringe | clear fringe |
| No clear fringe | 6 | 1 |
| clear fringe | 0 | 6 |
| Accuracy (%) | 92.3 | |

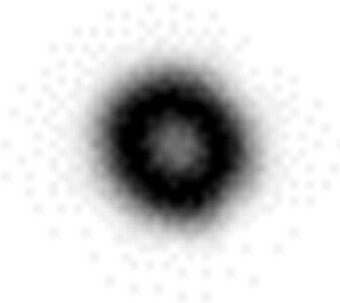

**Fig 18. Fringe pattern of samples with wrong predicted values.**

## 4 Conclusion

This study focused on uniform materials. By using FEA simulation and interpolation calculation, the fringe pattern containing phase information in ESPI technology was successfully simulated. The relationship between fringe characteristics and defect size, depth, and loading mode and loading intensity was explored. By establishing the simulation model, the influence of each variable on the defect detection and the corresponding laws are explored, and the loading mode and loading range applicable to the internal defect detection of aluminum materials are found out. The established simulation model and the obtained simulation results can provide theoretical guidance for the actual measurement of internal defects by the electronic speckle pattern interferometry system and improve the measurement accuracy. In actual experiments, complex factors such as wavelength effect, material anisotropy and multi-physics field coupling need to be comprehensively considered. Therefore, the absolute values measured in actual experiments may deviate from the simulation results. In future research, these aspects need to be further explored.

## Supporting information

**S1 Dataset. Minimal data set.**
(ZIP)

## Author contributions

**Data curation:** Mingzhi Lv.

**Methodology:** Wen Wang, Wenheng Li.

**Project administration:** Fang Zhang.

**Validation:** Wenheng Li.

**Writing – original draft:** Wen Wang.

**Writing – review & editing:** Fang Zhang, Zhitao Xiao.

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
