## [Decision Letter · Decision Letter 0]

Dear Dr. Zhang,

Thank you for submitting your manuscript to PLOS ONE. After careful consideration, we feel that it has merit but does not fully meet PLOS ONE’s publication criteria as it currently stands. Therefore, we invite you to submit a revised version of the manuscript that addresses the points raised during the review process.

We look forward to receiving your revised manuscript.

Kind regards,

Uma Maheswari Rajagopalan, Ph.D

Academic Editor

PLOS ONE

“This work was supported by Tianjin Metrology Technology Project (2024TJMT037).”

Reviewers' comments:

Reviewer's Responses to Questions

**Comments to the Author**

1. Is the manuscript technically sound, and do the data support the conclusions?

Reviewer #1: Yes

Reviewer #2: Partly

2. Has the statistical analysis been performed appropriately and rigorously?

Reviewer #1: N/A

Reviewer #2: N/A

3. Have the authors made all data underlying the findings in their manuscript fully available?

Reviewer #1: Yes

Reviewer #2: No

4. Is the manuscript presented in an intelligible fashion and written in standard English?

Reviewer #1: Yes

Reviewer #2: Yes

Reviewer #1: The study investigates the use of Electronic Speckle Pattern Interferometry (ESPI) in measuring internal defects within aluminum plates in relation to the effects of vacuum and thermal loading conditions. Using finite element analysis (FEA), the authors analyze the effects of defect size, depth, and loading parameters on interference fringe visibility, which is critical for flaw detection. The study suggests methods for reducing the loading conditions to enhance the accuracy of ESPI-based non-destructive testing.

The study is rigorous and relevant, but the following improvements would enhance its quality:

• The introduction could better emphasize the novelty of the work, such as the specific combination of FEA and ESPI for aluminum plates.

• Describe why specific defect size ranges (e.g., 20–60 mm) and depths (e.g., 0.3–0.9 mm) are being applied. Are they based on industrial standards?

• Provide more information on the interpolation method (PDE-based) and validation. How was the interpolation accuracy determined?

• Reference to possible limitations of the simulation, e.g. assumptions like laser wavelength effects. Is it possible to use such technique for composites material?

• Describe trends more forcefully (e.g., "Fringe density increases linearly with defect diameter").

• Is it possible to compare outcomes more directly with the earlier research (e.g., the work of Retheesh et al. on thermal loading)?

• The FE model assumes idealized boundary conditions and load transfer mechanisms, particularly specimen-loading device interaction (e.g., vacuum chamber or heat source). Surface roughness, contact, and pressure non-uniformity can actually influence fringe patterns and deformation response in real cases. How do the authors justify these assumptions? Was any sensitivity analysis done to assess how assumptions in the boundary conditions affect the simulated fields of displacement? A few lines on this deficiency and how this can affect detectability of flaws would strengthen the manuscript.

• Lack of experimental validation decreases practical application.

Reviewer #2: In this manuscript, simulations have been conducted to study defect inspection using ESPI. However, necessary experiments are missing, and the novelty is not clear. Detailed comments are given below.

1. The study is purely based on simulation. There are no experiments to support the conclusions.

2. What are the main novelty and contribution of the study? Some conclusions have been drawn on influencing parameters of ESPI. However, there is no in-depth analysis.

3. The workpiece being studied is simple aluminum plate. The defects inside can be easily detected by conventional radiographic testing, ultrasonic testing, eddy current testing etc. What is the motivation of using ESPI? What are the pros and cons of ESPI when compared with RT, UT and ECT?

4. Reference for eq.(1) should be added.

5. I suggest to use 2D view for fig.4, for better comparison with fig.5.

6. As shown in fig.5, the fringe pattern has great change after the interpolation. Why is the interpolation cause such a big change? How do you justify its correctness?

7. From the results, the fringe patterns are influenced by many parameters. What are the features that we could obtain from the pattern to evaluate the dimensions of defects.

**Do you want your identity to be public for this peer review?** For information about this choice, including consent withdrawal, please see our Privacy Policy

Reviewer #1: **Yes: ** Seyed Karen alavi

Reviewer #2: No

---

## [Author Response · Author response to Decision Letter 1]

6 Jun 2025

Dear reviewer:

Thank you for your decision and constructive comments on our manuscript. We have studied the comments carefully and have made revisions which marked in yellow in the revised manuscript. Revision notes, point-to-point, are given as follows.

Reviewer #1:

• Question: The introduction could better emphasize the novelty of the work, such as the specific combination of FEA and ESPI for aluminum plates.

Answer: Thank you for your revision suggestion. I have added a description of the innovation points and contributions of this article in the penultimate paragraph of the introduction. The main contributions of this article are mainly the following two points: (1) The out-of-plane displacement of the defect position after vacuum loading and thermal loading is simulated by FEA, and then the phase is calculated by the displacement to obtain the ESPI stripe. By studying the detection ability of different loading amounts for defects with different diameters and depths, an important basis can be provided for how to select loading mode and parameter in ESPI nondestructive testing and evaluate the defect detection capability of this testing technology. (2) The full-field displacement values are obtained by applying the partial differential equation (PDE)-based interpolation method to the discrete displacement data from the finite element mesh simulation. This PDE-based approach enables accurate interpolation of the displacement field across the entire domain.

• Question: Describe why specific defect size ranges (e.g., 20–60 mm) and depths (e.g., 0.3–0.9 mm) are being applied. Are they based on industrial standards?

Answer Thank you for your revision suggestion. This manuscript is a preliminary attempt to explore the boundary conditions of ESPI technology in its application through finite element simulation. Therefore, defects with large diameters ranging from 20.0mm to 60.0mm and small diameters ranging from 5mm to 10.0mm were simulated. This type of defect often occurs in the manufacturing of large components (such as ships and Bridges) or in the damage of complex service environments (such as deep sea and high temperature). The depths were all within 1mm to verify the detection force of ESPI technology for near-surface damage. The explanation of this part is supplemented to the first paragraph of Part 3 of the manuscript.

• Question: Provide more information on the interpolation method (PDE-based) and validation. How was the interpolation accuracy determined?

Answer Thank you for your revision suggestion. To demonstrate the effectiveness of the interpolation method, we present the following experimental results. Figure (a) shows the full field phase obtained through computer simulation (phase is proportional to out of plane displacement). Figure (b) shows the result of random sampling of Figure (a), where 15% of the phase values are retained and 85% of the phase values are discarded (set to zero). Figure (c) shows the full field phase recovered using the partial differential equation interpolation method proposed in this paper for processing Figure (b). Figures (d), (e), and (f) are three-dimensional visualizations of Figures (a), (b), and (c), respectively. From the comparison results of figures (a) and (c), as well as figures (d) and (f), it can be seen that the interpolation results based on partial differential equations have little difference from the true phase values. Actual calculations show that the average relative error between the interpolated full field phase and the true phase is 0.16%.

• Question: Reference to possible limitations of the simulation, e.g. assumptions like laser wavelength effects. Is it possible to use such technique for composites material?

Answer: Thank you for your revision suggestion. The sensitivity of ESPI is directly related to the laser wavelength. According to the interference principle, the sensitivity of displacement measurement is inversely proportional to the wavelength. That is, the shorter the wavelength, the smaller the displacement corresponding to the same phase change. The visible light band (such as 532 nm green light or 632.8 nm He-Ne laser) is a common choice for ESPI because it strikes a balance between sensitivity and device maturity.

ESPI technology has significant potential in the application of composite materials, and its simulation needs to comprehensively consider complex factors such as wavelength effect, material anisotropy and multi-physics field coupling. In practical applications, it is necessary to calibrate the simulation parameters (such as speckle size and filtering algorithm) through experiments to deal with the diversity of composite materials. Existing studies have shown that ESPI has achieved initial success in the vibration, failure detection and parameter reverse calculation of composite materials, such as the application mentioned in Reference [4].

• Question: Describe trends more forcefully (e.g., "Fringe density increases linearly with defect diameter").

Answer: Thank you for your revision suggestion. ESPI technology characterizes displacement or strain by analyzing the changes in speckle interference fringes on the surface of an object after it is subjected to force. The fringe density reflects the spatial rate of change of the displacement field, that is, the displacement gradient. According to the principle of phase measurement, the fringe density is directly proportional to the displacement gradient. When the loading force increases, the displacement gradient (strain) on the material surface increases, resulting in a decrease in the spacing between adjacent interference fringes, thereby increasing the fringe density. If the loading amount remains unchanged and the defect diameter increases, it will lead to a more likely surface displacement, thereby increasing the fringe density. Therefore, the fringe density is positively correlated with the loading amount and the defect diameter, but negatively correlated with the defect depth.This is further proved by the calculation results of regression coefficients in the newly added regression model in Section 3.3.

I've added a stronger explanation of this trend in sections 3.1.1 and 3.1.2

• Question: Is it possible to compare outcomes more directly with the earlier research (e.g., the work of Retheesh et al. on thermal loading)?

Answer: Thank you for your valuable suggestion. I have conducted a more direct and in-depth analysis and evaluation of the references mentioned in the introduction. Due to the fact that existing methods only provide finite element simulation results, rather than full field displacement, we did not compare with them.

To demonstrate the effectiveness of the interpolation method, we present the following experimental results. Figure (a) shows the full field phase obtained through computer simulation (phase is proportional to out of plane displacement). Figure (b) shows the result of random sampling of Figure (a), where 15% of the phase values are retained and 85% of the phase values are discarded (set to zero). Figure (c) shows the full field phase recovered using the partial differential equation interpolation method proposed in this paper for processing Figure (b). Figures (d), (e), and (f) are three-dimensional visualizations of Figures (a), (b), and (c), respectively. From the comparison results of figures (a) and (c), as well as figures (d) and (f), it can be seen that the interpolation results based on partial differential equations have little difference from the true phase values. Actual calculations show that the average relative error between the interpolated full field phase and the true phase is 0.16%.

• Question: Thank you for your revision suggestion. The FE model assumes idealized boundary conditions and load transfer mechanisms, particularly specimen-loading device interaction (e.g., vacuum chamber or heat source). Surface roughness, contact, and pressure non-uniformity can actually influence fringe patterns and deformation response in real cases. How do the authors justify these assumptions? Was any sensitivity analysis done to assess how assumptions in the boundary conditions affect the simulated fields of displacement? A few lines on this deficiency and how this can affect detectability of flaws would strengthen the manuscript.

Answer: Thank you for your revision suggestion. This study focused on uniform materials. By using FEA simulation and interpolation calculation, the fringe pattern containing phase information in ESPI technology was successfully simulated. The relationship between fringe characteristics and defect size, depth, and loading amount was explored. Eventually, it was proved that the simulation experiment results were consistent with the theoretical conclusions. In actual experiments, complex factors such as wavelength effect, material anisotropy and multi-physics field coupling need to be comprehensively considered. Therefore, the absolute values measured in actual experiments may deviate from the simulation results. In future research, these aspects need to be further explored.

I have supplemented this note in the conclusion of the manuscript.

• Question: Lack of experimental validation decreases practical application.

Answer: Thank you for your revision suggestion. In this manuscript, for the uniform aluminum plate material, by using FEA simulation and interpolation calculation, the fringe pattern containing phase information in ESPI technology was successfully simulated. The relationship between the fringe characteristics and the defect size, depth, and loading amount was explored. Finally, it was proved that the simulation experiment results were consistent with the theoretical conclusions, verifying the potential of the simulation experiment in the research of this topic. As mentioned above, in actual experiments, environmental factors such as wavelength effect, material anisotropy, and optical path construction, as well as noise factors in speckle interference, need to be comprehensively considered. It is predicted that the absolute values measured in actual experiments will deviate from the simulation results, and calibration through algorithms is required. The experimental verification related to the aluminum plate involved in this article has not been carried out yet, but it will be the key research direction for us in the future. Thank you for your guidance and suggestions.

Reviewer #2:

Question: 1. The study is purely based on simulation. There are no experiments to support the conclusions.

Answer: Thank you for your revision suggestion. In this manuscript, for the uniform aluminum plate material, by using FEA simulation and interpolation calculation, the fringe pattern containing phase information in ESPI technology was successfully simulated. The relationship between the fringe characteristics and the defect size, depth, and loading amount was explored. Finally, it was proved that the simulation experiment results were consistent with the theoretical conclusions, verifying the potential of the simulation experiment in the research of this topic. In actual experiments, environmental factors such as wavelength effect, material anisotropy, and optical path construction, as well as noise factors in speckle interference, need to be comprehensively considered. Based on the above considerations, it is expected that the absolute value of displacement measured in actual experiments may deviate from the simulation results to some extent and need to be calibrated through algorithms. The experimental verification related to the aluminum plate involved in this manuscript has not been carried out yet, but it will be the key research direction for us in the future. Thank you for your guidance and suggestions.

Question: 2. What are the main novelty and contribution of the study? Some conclusions have been drawn on influencing parameters of ESPI. However, there is no in-depth analysis.

Answer: Thank you for your revision suggestion. The main contributions of this article are mainly the following two points: (1) The out-of-plane displacement of the defect position after vacuum loading and thermal loading is simulated by FEA, and then the phase is calculated by the displacement to obtain the ESPI stripe. By studying the detection ability of different loading amounts for defects with different diameters and depths, an important basis can be provided for how to select loading mode and parameter in ESPI nondestructive testing and evaluate the defect detection capability of this testing technology. (2) The full-field displacement values are obtained by applying the partial differential equation (PDE)-based interpolation method to the discrete displacement data from the finite element mesh simulation. This PDE-based approach enables accurate interpolation of the displacement field across the entire domain.

We have added a description of the innovation points and contributions of this manuscript in the penultimate paragraph of the introduction.

On the other hand, I added new section “3.3 Qualitative Analysis and Prediction of Defect Detection Based on Logistic Regression” as a further in-depth analysis and application of the measured data.

Question: 3. The workpiece being studied is simple aluminum plate. The defects inside can be easily detected by conventional radiographic testing, ultrasonic testing, eddy current testing etc. What is the motivation of using ESPI? What are the pros and cons of ESPI when compared with RT, UT and ECT?

Answer: Thank you for your revision suggestion. While conventional methods (RT, UT, ECT) effectively detect internal defects in aluminum, ESPI (Electronic Speckle Pattern Interferometry) offers unique advantages in specific scenarios:1. Real-time full-field monitoring: ESPI can provide a global view of the surface deformation of the workpiece, and analyze the dynamic changes in strain caused by defects under loads (such as mechanical or thermal stress) in real time, which is superior to the point scanning mode of UT/ECT. 2. Non-contact and non-destructive: No physical probes or coupling agents are required as in the UT method, nor is there the radiation risk as in the RT method, making it safer for operators.3. High sensitivity: detecting micron-scale displacement, revealing subtle defects or residual stresses that may not significantly change the internal structure but affect the surface behavior, good at detecting near-surface defects, complementary to RT/UT. On the other hand, for some kind of materials, such as rubber material, this method has obvious advantages.

I have added relevant statements in the second paragraph of the introduction and thank you for your comments.

Question: 4. Reference for eq.(1) should be added.

Answer: Thank you for your revision suggestion. The reference for eq.(1) has been supplemented, which is reference 10.

Question: 5. I suggest to use 2D view for fig.4, for better comparison with fig.5.

Answer: Thank you for your suggestion. I have supplemented the two-dimensional view of the out-of-plane displacement in the manuscript, as shown in Figure 4. To observe the change of the out-of-plane displacement value more obviously, I have also retained its three-dimensional view, as shown in Figure 5.

Question: 6. As shown in fig.5, the fringe pattern has great change after the interpolation. Why is the interpolation cause such a big change? How do you justify its correctness?

Answer Thank you for your revision suggestion. The generation of ESPI fringes depends on the continuity of phase information, while the out-of-plane displacement values output by finite element simulation are usually discrete grid node values (as shown in Fig. 3). At the junctions of adjacent grid cells, the displacement data may change abruptly, resulting in discontinuous stripes or obvious grid division traces (similar to a "pixelation" effect). This change demonstrates the effectiveness of the interpolation method based on partial differential equations used in this article. This method considers the displacement of known points as a continuous heat source, and fills the displacement of missing points through energy diffusion, ultimately achieving a stable state of heat distribution, that is, obtaining the full f

---

## [Decision Letter · Decision Letter 1]

The modeling and condition analysis of nondestructive testing based on ESPI for internal defects of materials

PONE-D-25-16310R1

Dear Dr. Zhang,

We’re pleased to inform you that your manuscript has been judged scientifically suitable for publication and will be formally accepted for publication once it meets all outstanding technical requirements.

Kind regards,

Uma Maheswari Rajagopalan, Ph.D

Academic Editor

PLOS ONE

Additional Editor Comments (optional):

Reviewers' comments:

Reviewer's Responses to Questions

**Comments to the Author**

Reviewer #1: All comments have been addressed

2. Is the manuscript technically sound, and do the data support the conclusions?

Reviewer #1: Partly

3. Has the statistical analysis been performed appropriately and rigorously?

Reviewer #1: Yes

4. Have the authors made all data underlying the findings in their manuscript fully available?

Reviewer #1: Yes

5. Is the manuscript presented in an intelligible fashion and written in standard English?

Reviewer #1: Yes

Reviewer #1: (No Response)

**Do you want your identity to be public for this peer review?** For information about this choice, including consent withdrawal, please see our Privacy Policy

Reviewer #1: No

---

## [Editor Report · Acceptance letter]

PONE-D-25-16310R1

PLOS ONE

Dear Dr. Zhang,

I'm pleased to inform you that your manuscript has been deemed suitable for publication in PLOS ONE. Congratulations! Your manuscript is now being handed over to our production team.

Kind regards,

on behalf of

Dr. Uma Maheswari Rajagopalan

Academic Editor

PLOS ONE